# The Effectiveness of the Low-Glycemic and Insulinemic (LOGI) Regimen in Maintaining the Benefits of the VLCKD in Fibromyalgia Patients

**DOI:** 10.3390/nu16234161

**Published:** 2024-11-30

**Authors:** Giuseppe Castaldo, Carmen Marino, Maria D’Elia, Manuela Grimaldi, Enza Napolitano, Anna Maria D’Ursi, Luca Rastrelli

**Affiliations:** 1NutriKeto_LAB Unisa, “San Giuseppe Moscati” National Hospital (AORN), Contrada Amoretta, 83100 Avellino, AV, Italy; giuseppecastaldo@yahoo.it; 2Department of Pharmacy, University of Salerno, Via Giovanni Paolo II 132, 84084 Fisciano, SA, Italy; cmarino@unisa.it (C.M.); or maria.delia01@unipa.it (M.D.); magrimaldi@unisa.it (M.G.); enapolitano@unisa.it (E.N.); 3NBFC, National Biodiversity Future Center, 90133 Palermo, SI, Italy; 4Department of Earth and Marine Science, University of Palermo, 90127 Palermo, SI, Italy

**Keywords:** fibromyalgia, ^1^H NMR metabolomics,VLCKD diet, LOGI diet

## Abstract

**Background:** Fibromyalgia (FM) is a chronic disorder that causes damage to the neuro-muscular system and alterations in the intestinal microbiota and affects the psychological state of the patient. In our previous study, we showed that 22 women patients subjected to a specific very low-carbohydrate ketogenic therapy (VLCKD) showed an improvement in clinical scores as well as neurotransmission-related and psychological dysfunctions and intestinal dysbiosis. Furthermore, NMR metabolomic data showed that changes induced by VLCKD treatment were evident in all metabolic pathways related to fibromyalgia biomarkers. **Methods**: Based on this evidence, we extend our investigation into dietary interventions for fibromyalgia by evaluating the impact of transitioning from a VLCKD to a low-glycemic insulinemic (LOGI) diet over an additional 45-day period. Therefore, participants initially following a VLCKD were transitioned to the LOGI diet after 45 days to determine whether the improvements in FM symptoms and metabolic dysfunctions achieved through VLCKD could be sustained with LOGI. **Results**: Our findings suggested that while VLCKD serves as an effective initial intervention for correcting metabolic imbalances and alleviating FM symptoms, transitioning to a LOGI diet offers a practical and sustainable dietary strategy. This transition preserves clinical improvements and supports long-term adherence and quality of life, underscoring the importance of adaptable nutritional therapies in chronic disease management. Control patients who adhered only to the LOGI diet for 90 days showed only modest improvement in clinical and psychological conditions, but not elimination of fibromyalgia symptoms. **Conclusions**: In conclusion the LOGI diet is an excellent alternative to maintain the results obtained from the regime VLCKD.

## 1. Introduction

Fibromyalgia (FM) is a chronic disorder characterized by widespread neuromuscular pain, mood disturbances, and alterations in gut microbiota, collectively referred to as intestinal dysbiosis [1,2,3,4,5,6,7]. Despite extensive research efforts, the underlying etiology of FM remains inadequately understood [1,3,8,9].

FM is diagnosed considering the patient’s clinical history and the patient’s responses to the Symptom Severity Score (SSS) and Widespread Pain Index (WPI)) [1,3]. These parameters are not objective, making the diagnosis difficult and uncertain, especially in the presence of other recurrent comorbidities. Pharmacological therapy is based on relaxants, antidepressants, and non-steroidal anti-inflammatory drugs [10,11,12,13].

However, these medications are symptomatic and do not address the underlying causes of the pathology, thus offering a treatment that does not lead to a cure for the disease.

Recent studies suggest a potential link between FM and systemic metabolic dysfunctions with a shared inflammatory origin [14,15,16,17,18]. As a proof of concept, we previously demonstrated that the metabolome of fibromyalgia patients exhibits a significant decrease in glucuronic acid, a toll-like receptor *4* ligand that exacerbates inflammatory conditions and increases pain severity [5,19].

The systemic nature of the disease, the lack of a gold-standard treatment, and the palliative effects of many pharmacological options have prompted the consideration of diet therapy as a potential alternative [15,20]. Indeed, therapeutic interventions based on diet therapy have been proven effective in many diseases [16,17,21,22,23]. Our previous research investigated the use of VLCKD in psoriasis patients [18,24]. In this case, we demonstrated that the catabolism of visceral fat, which in those patients produces inflammatory kinins, was dramatically reduced. The VLCKD diet is a nutritional regimen characterized by a drastic reduction in carbohydrates [25], where the paucity or absence of carbohydrates causes an increase in blood ketones, shifting the energetic metabolism towards the body’s ketone metabolism.

Other than obesity, nutritional approaches based on VLCKD proved effective in treating metabolic syndrome (diabetes), neurological and autoimmune diseases, acne, polycystic ovary syndrome, and cancer [21,24,26,27,28]. VLCKD improved symptoms and corrected several out-of-range laboratory parameters thanks to the rebalancing of systemic metabolic dysfunctions [29]. Besides exploring diet-based interventions to cure psoriasis [18,24,30], our research has also focused on the efficacy of calcium-rich mineral water as a beneficial alternative to caloric Ca-rich dairy products in addressing Ca dysmetabolism. Using comprehensive laboratory tests, including NMR-based metabolomic analyses of urine and blood sera, we demonstrated that consuming 2 L of bicarbonate-calcium mineral water (Lete^®^) daily over six months was effective in rebalancing Ca dysmetabolism and preserving bone integrity [31].

We recently demonstrated that 41 female FM patients subjected to a specific VLCKD, i.e., the oloproteic diet, exhibited improved FM symptoms and enhanced systemic metabolic conditions. Based on NMR metabolomic data, changes induced by the VLCKD treatment were evident in all the metabolic pathways related to fibromyalgia dysfunction [19], in particular, (i) rebalancing of inflammatory conditions [32,33,34,35]; (ii) up-regulation of catecholamine transmission; (iii) modulation of GABAergic transmission; (iv) availability of additional energy sources to cope with neuromuscular stress conditions [36,37,38,39].

Despite the effectiveness of the VLCKD, the long-term sustainability of the ketogenic diet remains a concern due to its restrictive nature.

The present study extends our investigation into dietary interventions for FM by evaluating the impact of transitioning from a VLCKD to a low-glycemic insulinemic (LOGI) diet over an additional 45-day period, from day 45 to day 90. Therefore, in this paper, our objective was to assess the effects of discontinuing VLCKD and adopting a LOGI, a more sustainable dietary approach, on FM patients’ clinical parameters and metabolic profiles [40]. The LOGI diet is a nutritional regimen that includes moderate low-glycemic index carbohydrate content, supplemented by a significant amount of protein fulfilling the dietary standard values, and a fat intake adjusted based on the patient’s BMI. The LOGI diet’s nutrient distribution aligns with the foundational principles of the Mediterranean diet in its original conception, encouraging the consumption of fish, vegetables, and whole grains to maintain a controlled glycemic response and provide a more flexible and palatable nutritional regimen than VLCKD [40].

In this study, participants initially following a VLCKD were transitioned to the LOGI diet after 45 days to determine whether the improvements in FM symptoms and metabolic dysfunctions achieved through VLCKD could be sustained with LOGI, thereby providing a viable long-term nutritional strategy for FM management.

By employing a combination of clinical evaluations and non-invasive NMR metabolomic analyses of serum, urine, and saliva samples, we monitored the physiological and biochemical changes in the participants from day 45 to day 90 [41].

NMR spectroscopy represents a robust and suitable technique for metabolomic studies, enabling the simultaneous qualitative and quantitative identification of low-molecular mass compounds in biological fluids [5,42].

Our findings suggest that while the VLCKD diet serves as an effective initial intervention for correcting metabolic imbalances and alleviating FM symptoms, transitioning to a LOGI diet offers a practical and sustainable dietary strategy. This transition preserves clinical improvements and supports long-term adherence and quality of life, underscoring the importance of adaptable nutritional therapies in chronic disease management. Patients following the LOGI diet for all 90 days exhibited a modest enhancement in their clinical condition but did not eliminate symptoms of fibromyalgia.

## 2. Materials and Methods

### 2.1. Participants and Sample Size Calculation

Sample size calculation was performed using G power 3.1.9.2 software (Softonic, Düsseldorf, Germany) [43], as reported by Castaldo et al. [19]. Accordingly, 50 FM patients were recruited based on rheumatologist diagnosis using the ACR criteria 2010 [3]. The eligibility criteria are reported in Table 1. A dropout of nine patients was reported during the study. As previously reported in ref. [21], patients were classified into (i) the FM1 group, following the VLCKD diet for the first 45 days and then the LOGI regimen, and (ii) the FM2 group, following the LOGI diet for all 90 days. One patient dropped out in the FM1 group and four dropped out in the FM2 group in the t45–t90 time range. The clinical biochemical evaluation was made to validate the inclusion and exclusion criteria (Table 1)

Demographic characteristics are reported in Table 2.

### 2.2. Clinical and Laboratory Evaluation

Clinical and laboratory evaluation was performed as previously reported [18,19,21]. Each patient in the FM1 and FM2 groups underwent anamnestic and rheumatological clinical evaluation at t0 and t45. The patients were followed regarding the standard FM symptoms to evaluate their protocol compliance and symptoms/benefits.

In detail, biochemical analyses of urine, serum, and nutritional assessments were carried out at times 0, 45, and 90. The results are reported in Table 1. In parallel, clinical evaluations and metabolomic analyses were performed.

The rheumatological assessment was carried out according to the ACR 2010 criteria. Patients were invited to give a semi-quantitative estimation of FM comorbidities on a scale of 0 to 3 (0: symptom absence; 1: low-intensity symptom; 2: medium intensity; 3: high intensity).

The rheumatological assessment was repeated at t0 and t45 for all the patients, as the rheumatologists were unaware of the nutritional interventions provided to patients.

Clinical evaluation of anxiety and depression was performed using the Hamilton Anxiety Rating Scale and Hamilton Depression Rating Scale (HAM-A and HAM-D); a pathological state was associated with HAM-A > 17 and HAM-D > 21 [5].

### 2.3. Dietary Intervention and Assessment

The VLCKD was inspired by the Blackburn diet (PSMF—protein-sparing modified fast) [44]. It is a non-carbohydrate diet with around 10 g of carbohydrates per day in vegetables and foods such as yogurt. The protein content was equal to 1.4 g/kg/ideal body weight calculated according to the Lanzola formula [45]. The protein quota was divided between non-glucidic and low-fat protein foods such as meat and fish. The patients took hydrolyzed whey protein powder supplements with added amino acids to avoid the loss of lean mass and stimulate GH synthesis. The amino acid composition of the protein powder included AKG (arginine keto glutarate), OKG (ornithine keto glutarate), and 10 g of hydrolyzed marine or porcine collagen. The lipid quota was variable (from 45 to 100 g), as it was inversely proportional to the BMI and was made up of monounsaturated lipids and MCTs (extra virgin olive oil and coconut oil). Abundant quantities of cooked or raw vegetables were provided for lunch and dinner, excluding those containing more than 1.5 g of carbohydrates per 100 g of edible material. To make the diet safe and effective, the protocol included (i) uricosuric phytotherapeutic drains (orthosiphon and taraxacum) with a natriuretic effect and reduced potassium excretion; (ii) alkalizing salts based on potassium, magnesium, and calcium carbonates and citrates; (iii) and seasoning salts including 2/3 potassium chloride and citrate and 1/3 sodium chloride. A total of 2 L per day of bicarbonate-alkaline water was prescribed to buffer the acidogenic activity of the ketone bodies. Intestinal draining Ayurvedic herbs such as Triphala were prescribed to prevent constipation, often recurrent in ketogenic regimes. We excluded drastic laxatives or purgatives to avoid significant losses of potassium. Alcoholic drinks were prohibited and stimulant drinks such as coffee or tea were limited. At enrolment, all treatments with oral hypoglycemics and diuretics were discontinued.

The LOGI diet is a balanced Mediterranean-inspired diet characterized by the following elements:

(i) The prescribed calorie content varied from 20 to 35 Kcal/kg of ideal body weight calculated according to the Lanzola formula, with a reduced calorie intake for patients with a BMI > 25 and a higher calorie content for patients with a lower BMI. (ii) The prescribed breakfast was essentially protein-based to keep insulin levels low in the morning and better modulate the sense of hunger. (iii) The general carbohydrate quota prescribed was less than 50% calories, made up of legumes and carbohydrates with a low glycemic index (legumes, quinoa, rice, corn, buckwheat, including pasta only once a week and unleavened cereal biscuits without gluten), thus excluding or limiting gluten and lactose, with moderate quantities of fresh seasonal fruit and abundant quantities of vegetables at both lunch and dinner, and excluding leavened foods. The average quantity prescribed was approximately 150 g of carbohydrates. (iv) The protein quota was 1.2 g per kg of ideal body weight calculated with the Lanzola formula, including protein foods such as meat, fish, eggs, Greek yogurt, and parmesan cheese, and 10 g of hydrolyzed marine or porcine collagen. (v) The prescribed lipid quota, including monounsaturated lipids and MCTs (extra virgin olive oil and coconut oil), varied from 45 to 100 g and was inversely proportional to the patients’ BMI.

Patients were advised to drink at least 2 L/day of low-sodium mineral water. Alcoholic drinks were banned, and stimulating drinks such as coffee and tea were limited.

### 2.4. Sample Preparation for NMR Metabolomic Analysis

As previously reported in refs. [5,19,41], all samples were prepared according to standard operating procedures (SOP). Serum was collected using tubes without anticoagulants. The serum was centrifugated at 12,000× *g* before NMR spectroscopy acquisition. After centrifugation at 12,000× *g*, the blood serum was aliquoted and stored at −80 °C in Greiner cryogenic vials before NMR spectroscopy measurements. NMR tubes were prepared using 200 μL of phosphate buffer, including 0.075 M Na_2_HPO4 ×7 H_2_O, 4% NaN_3_, and H_2_O and 300 μL of blood sera. Trimethylsilyl propionic-2,2,3,3-d4 acid (TSP 0.1% in D_2_O) was used as an internal reference for the alignment and quantification of NMR signals. The mixture was transferred to a 5 mm NMR tube (Bruker NMR tubes) before analysis started. According to ref. [46], 1.5 mL of 24 h urine was centrifuged at 15.000× *g* for 10 min and then filtered using a 0.2 µm filter. Finally, 500 μL of urine was transferred into a new tube and added to 50 μL of 50 mM phosphate buffer in 99.8% D_2_O [47,48].

Saliva samples were collected using Sartstedt Salivette^®^, Numbrect, Germany. After centrifugation, samples were transferred to a 5 mm heavy-walled NMR tube. In detail, 425 µL of each saliva sample was added to 25 µL of 1 M potassium phosphate buffer (pH 7.4) and 10 µL D_2_O. TSP 0.1% in D_2_O was used as an internal reference for aligning and quantifying the NMR signals.

### 2.5. NMR Data Acquisition and Processing

A Bruker Ascend™ 600 MHz spectrometer equipped with a 5 mm triple resonance Z gradient TXI probe (Bruker Co., Rheinstetten, Germany) was used at 298 K. TopSpin version 3.2 was used for NMR data acquisition (Bruker Biospin). CPMG (Carr-Purcell–Meiboom–Gill) experiments were performed on serum and urine samples using a 20 ppm spectral width, 32 k data points, f1 presaturation, and a T2 filter using a D20 of 300 µs and D1 of 4 s. A weighted Fourier transform was applied to the time domain data with a line width of 0.5 Hz, followed by a manual step and baseline correction in preparation for targeted profiling analysis. As previously reported, R package ASICS was used to identify and quantify biofluids’ metabolites [19,31]. The package combines all the phases of the analysis (management of a reference library with spectra of pure metabolites, pre-processing, quantification, diagnostic tools to evaluate the quality of quantification, and post-quantification statistical analysis) [49].

### 2.6. Statistical Analysis

The statistical analysis of clinical parameters and pathological index scores was conducted using an ANOVA one-way test with post hoc correction. Moreover, a heatmap was constructed using the Euclidean distance and Ward method to assess the similarity of clinical profiles in the two groups at the three time points (t0, t45, t90) [49]. Normalized concentration matrices were created by supervised multivariate statistical approaches applying PLS-DA using MetaboAnalyst 6.0. The performance of the PLS-DA model was evaluated using the coefficient Q2 (using the 7-fold internal cross-validation method) and the coefficient R2, which define the variance predicted and explained by the model, respectively [50]. The significant metabolites responsible for maximum separation in the PLS-DA were ranked according to their variable influence on projection (VIP) scores, considering only the metabolites with VIP >1 to be significant. The enrichment analysis tool used Metaboanalyst 6.0 to predict the dysregulated biochemical pathways (Pang, Chong, Zhou, de Lima Morais). We considered significant only the biochemical pathways with hits values >2 and *p*-values < 0.001.

## 3. Results

### 3.1. Clinical Analysis

FM1 patients subjected to the VLCKD diet for 45 days (t0–t45) were invited by our nutritionists to follow a LOGI-based nutritional regimen for an additional 45 days (t45–t90). Conversely, FM2 patients enrolled to follow the LOGI diet from t0 to t45 were invited to continue their LOGI nutritional regimen from t45 to t90

Table 3 and violin graphs in Figure 1 show the fibromyalgia Widespread Pain Index (WPI) and Symptom Severity Scale (SSS) clinical parameter scores for FM1 and FM2 patients at t0, t45, and t90. HAM-A and HAM-D psychological parameters for both the FM patients are also shown. A one-way ANOVA test with post-hoc correction was performed to evaluate the reliability of the clinical and psychological parameters at the time points and the relative differences.

The comparison of WPI and SSS score values in FM1 patients from t0 to t90 indicates a 51.48% WPI and 33.27% SSS score reduction, respectively. As reported previously by our research group, a significant decrease in the WPI and SSS scores is evident from t0 to t45 (63.15% and 42.7%) [19]. Conversely, the downward trend of the WPI and SSS scores slowed down in the t45–t90 second-time interval when fibromyalgia patients followed the LOGI diet protocol. WPI and SSS scores for FM2 patients resulted in 25.00% and 24.15% reductions at t90 compared to t0. This reduction in WPI and SSS scores at t90 for FM2 patients mirrors the changes observed at t45, indicating that the LOGI diet maintains its moderate beneficial effects over time. (Figure 1).

It is well known that fibromyalgia is associated with anxiety and depression. We previously demonstrated that the severity of depression worsens dysmetabolic fibromyalgia conditions. Anxiety and depression are diagnosed using the Hamilton Anxiety and Depression Tests (HAM-A and HAM-D); a pathological state is defined by HAM-A scores above 17 and HAM-D scores above 21.

Table 2 and the graphs in Figure 1 show that FM1 patients at t90 experienced 55.50% and 47.22% reductions in HAM-A and HAM-D scores, respectively, compared to t0. By comparing HAM-A and HAM-D scores of FM1 patients at t90 with respect to t45, an 8.28% increase is evident. Even in this case, we can conclude that adopting a LOGI diet in the t45–t90 time range allows us to maintain the significant benefits of VLCKD observed from t0 to t45.

Observing anxiety and depression scores at t90 compared to t0 for FM2 patients, decreases of 25.37% and 3.17% for HAM-A and HAM-D, respectively, are evident. (Figure 1)

The data previously reported are represented in a heat map representation (Figure 2). The graph shows that the average clinical and psychological profiles of FM1 and FM2 patients before the dietary interventions (FM1_t0 and FM2_t0) are clustered together, indicating a similar starting clinical state. The parameters at t45 and t90 for FM1 patients are closely aligned and distinct from those at t0, suggesting consistent improvement. Similarly, the parameters for FM2 patients at t45 and t90 are close and similar to each other (light grey) and slightly different from those at t0.

### 3.2. Multivariate Data Analysis and Enrichment Analysis

To verify if clinical and psychological data trends correspond to similar tendencies in metabolomic data, the blood sera, urine, and saliva of FM1 and FM2 patients were analyzed at t90 using NMR-based metabolomic methods. The data matrices included 95 serum and 77 urine metabolite concentrations. After normalization according to the sum, log transformation, and Pareto scaling, the data matrices were analyzed by PLS-DA (Figure 3). The clustering validation was performed by calculating the Q2 and R2 index obtained with cross-validation methods (Appendix A). The data were compared to the metabolite concentrations measured at t0 and t45. As evident from the representation of the PLS-DA scatter plot in Figure 3, the multivariate statistical analysis of serum (Figure 3A) and urine (Figure 3B) metabolomic data indicates that the metabolomic profile of FM1 patients at t90 differs from those at t0 and t45. In contrast, the metabolomic profiles of FM2 patients at t0, t45, and t90 overlap, especially those at the t45 and t90 time points.

Table 4 reports the metabolites responsible for the specificity of the metabolomic fingerprints at t45 and t90. Notably, the metabolites responsible for the changes from t0 to t45 continue to play a critical role in the t45 to t90 interval. Many metabolites in the blood sera—D-glucuronic acid, ornithine, saccharic acid, phenylalanine, and malic acid—continue to increase or decrease from t45 to t90, as in the interval from t0 to t45, following the virtuous trend inducing the remission of symptoms. In contrast, the concentration of several metabolites in urine—valerate, isoleucine, serine, citrulline, TMAO, fucose, and beta-hydroxyisovalerate—from t45 to t90 varies oppositely to the t0–t45 time range. In some cases, this reversal is also evident for blood serum metabolites—guanidinoacetate, beta-hydroxyvalerate, glycogen—explaining the worsening of the clinical and psychological symptoms as previously reported. The enrichment approach evaluated the discriminating pathways between FM1_t45 and FM1_t90. (Table 5) The analysis confirmed the results obtained in the previous study, evidencing dysregulation of the amino acid pathways linked to neurotransmissions such as (i) glycine, serine, and threonine metabolism; (ii) alanine, aspartate, glutamate metabolism;(iii) tyrosine metabolism; and (iv) taurine and hypotaurine metabolism.

## 4. Discussion

In our previous work, we demonstrated the efficacy of the VLCKD diet in improving FM symptoms and the systemic metabolic conditions of fibromyalgia (FM) patients following 45 days of VLCKD treatment [19]. Despite the success of this nutritional regimen, its restrictive nature compromises long-term sustainability. In the present study, we evaluated a less restrictive low-glycemic insulinemic (LOGI) diet as a possible sustainable and somewhat enjoyable dietary regimen to retain the benefits obtained after 45 days of the VLCKD diet. We evaluated the trends of clinical, rheumatological, psychological, and metabolomic profile parameters over the period from day 45 to day 90 in FM1 and FM2 fibromyalgia patients who had previously followed the VLCKD diet and the LOGI diet, respectively, over a 45-day period (t0–t45).

The LOGI diet is a nutritional protocol designed according to the founding principles of the Mediterranean diet. It includes moderate low-glycemic index carbohydrates (vegetables and whole grains), a significant amount of protein (meat, fish, eggs) with respect to the standard nutritional values, and a fat intake according to the patient’s BMI [40].

Our data indicates that FM1 patients experienced a general net improvement in their clinical and psychological conditions at t90. However, clinical parameters slightly worsened if we compared t45 with the t90 data.

On the other hand, FM2 patients who followed the LOGI diet for 90 days experienced a modest improvement in clinical and psychological symptoms. These data prove that the LOGI diet, although not highly restrictive, is effective as a healthy nutritional regimen in controlling, although not resolving, the symptoms of the fibromyalgia inflammatory condition.

To test whether the change in clinical and psychological parameters corresponded to a change in systemic metabolism, we performed metabolomic analysis using NMR on serum, urine, and saliva. As previously demonstrated, the metabolome of both the serum and urine of FM1 patients changed significantly from t0 to t45. This work shows that 45 days of the LOGI diet after the VLCKD diet has an additional effect on the metabolome of FM1 patients. The same is not evident for FM2 patients, whose metabolomic profiles at t0, t45, and t90 are almost similar (overlapping in PLSDA scatter plots of Figure 3), thus explaining the modest remission of clinical symptoms and the undetectable change in the whole metabolic condition.

Interestingly, the metabolites that most significantly separate the metabolomes of FM1 patients in the t45–t90 time range are the same ones responsible for the change in the t0–t45 range. Notably, although the VLCKD protocol was an effective ketogenic regimen, including almost null carbohydrate content, the metabolites involved in modifying the metabolome were not exclusively those related to energetic metabolism.

Phenylalanine is confirmed to be reduced in the metabolome of FM1 patients in the t45–t90 time range, while isoleucine urinary concentration decreases in the t45–t90 time range, oppositely to t0–t45. Coherent with these data, enrichment analysis revealed modification of biochemical pathways related to neurotransmitter amino acids such as (i) alanine, aspartate, and glutamate; (ii) D-glutamine and D-glutamate; (iii) phenylalanine and tyrosine; and (iv) taurine and hypotaurine (Table 5). As previously demonstrated, the rebalancing of the neurotransmission amino acid metabolism explains the improvement of fibromyalgia symptoms such as neuropathic and muscular pain [51,52].

Moreover, the decrease in phenylalanine from t45 to t90 supports the up-regulation of catecholamine synthesis and transmission that significantly contributes to correcting an altered pain perception, typical of FM patients [53]. The reduction in phenylalanine is often correlated with an upregulation of the catecholamine pathway that has a positive impact on mood and muscle tone [53,54,55].

Glucuronic acid was confirmed to be decreased at t90 in FM1 patients (Table 4). This compound is a toll-like 4 receptor ligand with an inflammatory action almost responsible for pain severity [19,40,56]. The decrease in glucuronic acid indicates that the LOGI diet, in the t45–t90 time range, is effective in controlling inflammation at the origin of many fibromyalgia symptoms, particularly pain severity [40].

A significant decrease in dysbiosis biomarkers in urine (including hydroxy valerate, valerate, citrulline, and TMAO) and saliva (glycolate and urea) [57,58] (Table 4; Table 5) was confirmed in the t45–t90 time range for FM1 patients. Reductions in these metabolites are associated with improving inflammatory conditions in the gut. These metabolites are highly produced by the intestinal microbiota in dysbiosis conditions [59,60,61,62]. In particular, TMAO has been identified as one of the pro-inflammatory metabolites of the endothelium and intestinal villi [63].

Clinical data show that a very high proportion of FM patients manifest recurrent cystitis, vaginal discharge, and intestinal impairment symptoms [57,58]. Meteorism and irritable bowel syndrome (IBS), which are almost constant in FM symptomatology, suggest an important role of microbiota alteration in the etiology of FM. Accordingly, we previously found that a metabolomic condition associated with a regression of intestinal dysbiosis correlates with the genetic trehalose exchange enzyme switching-off involved in the Candida albicans cell wall synthesis (entry EC 2.3.1.122) [19]. This finding underlines the importance of a correct nutritional approach in coping with FM symptoms by controlling the alteration of intestinal microbiota and possibly C. albicans infections [19].

It is well known that fibromyalgia is associated with an alteration in GABAergic neurotransmission [32,64]. Indeed, several pharmacological therapeutic interventions make use of GABA inhibitors [65,66]. While the serum metabolome of FM1 patients at t45 exhibits significantly lowered concentrations of the GABAergic metabolites GABA and guanidinoacetate (GAA), this tendency reverses in the t45–t90 period, further supporting the reduction in symptom remission in this time range.

## 5. Conclusions

Taken altogether, our data show that

(i)The LOGI diet maintains the improvements achieved after 45 days of the VLCKD therapeutic regimen in FM1 intervention patients. By following an equilibrated and non-restrictive LOGI nutritional regimen, FM patients can sustain the remission of symptoms.(ii)The LOGI diet, kept for 90 days in FM2 patients, is inefficient in inducing the metabolic reset necessary to reverse the metabolic conditions associated with FM disease. However, the effect of LOGI, albeit modest, in reducing symptomatology and modifying the metabolome underlines the importance of a correct and clean diet regimen in avoiding the worsening of FM patients’ clinical and metabolic conditions.

## Figures and Tables

**Figure 1 nutrients-16-04161-f001:**
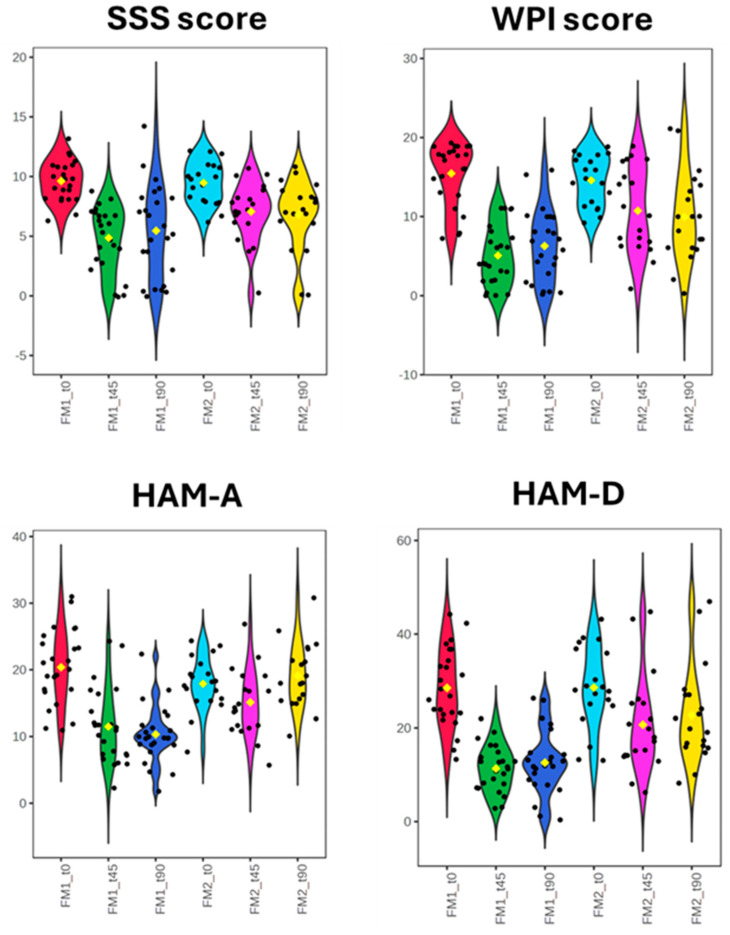
Violin graphs representing fibromyalgia patients’ rheumatological and psychological scores before and after nutritional treatment. The yellow diamond represents the average value.

**Figure 2 nutrients-16-04161-f002:**
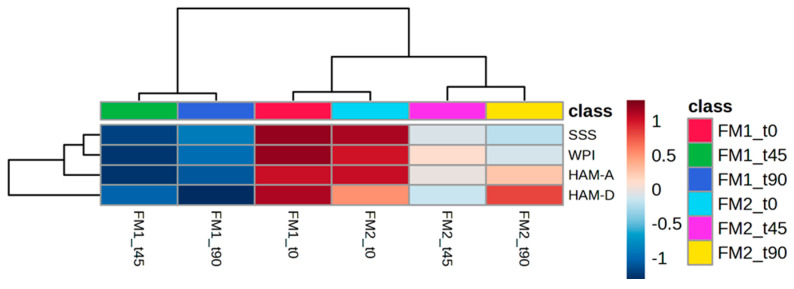
Heatmap of changed clinical scores. The colour of each section corresponds to the concentration value of each clinical score calculated by the rheumatological and psychiatric test (red, upregulated; blue, downregulated).

**Figure 3 nutrients-16-04161-f003:**
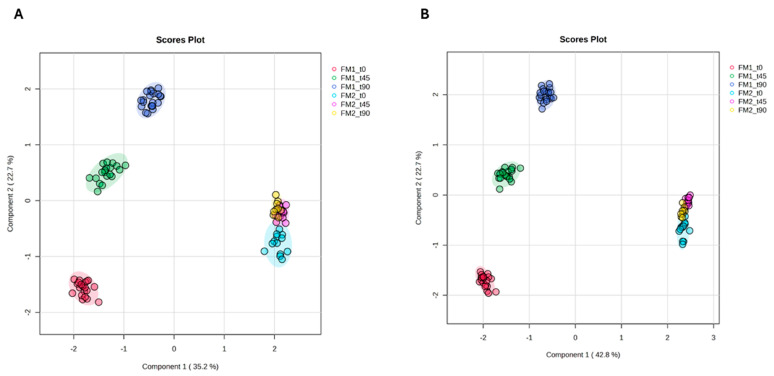
PLS-DA score scatter plot for the 1H NMR data collected in 1D-1H-CPMG spectra for serum and urine and acquired at 600 MHz. Data represent the sera (**A**) and urine (**B**) from fibromyalgia patients before (FM1_t0, FM2_t0: red and light blue clusters, respectively) and after 45 days of VLCKD (FM1_t45: green); after 45 days of the LOGI diet (FM2_t45: purple); and after an additional 45 days of the LOGI diet (FM1_t90; FM2_t90: blue and yellow clusters, respectively).

**Table 1 nutrients-16-04161-t001:** Inclusion and exclusion criteria.

Inclusion Criteria
Adult women older than 18 years until pre-menopause Diagnosis of FM performed by the rheumatologist according to the Rome III criteria of the American College of Rheumatology, revised in 2010 Ability to read and sign informed consent Normal kidney function with serum creatinine of 1.2 mg/dL and a glomerular filtration rate of 80 mL/min Normal liver function with ALT-AST-GGT
Exclusion criteria
Patients with pathologies that prevent following the ketogenetic dietary intervention Patients currently breastfeeding or pregnant Previous or current clinical history of drug or other substance abuse Presence of other inflammatory diseases Renal failure (creatinine: 1,4 mg/dL) or glomerular filtration rate lower than 80 mL/min Severe or medium-severe liver failure Insulin-dependent diabetes (Type 1) Ventricular atrium block with a QT of 0.44 ms Cardiac arrhythmias Severe or medium-severe heart failure Uncontrolled hypokalemia Persistent diarrhea or vomiting Heart attack or stroke in the last 12 months Ongoing neoplasms Serious psychiatric disorders

**Table 2 nutrients-16-04161-t002:** Demographic information.

Parameters	Fibromyalgia First Group (N = 22)	Fibromyalgia Second Group (N = 19)
Sex (male/female)	0/22	0/19
Age (mean ± SD, year)	42.66 ± 8	40.22 ± 3
BMI (kg/m^2^)	31.82 ± 2.66	27.36 ± 7.12
Smokers (YES/NO)	5/17	3/16

**Table 3 nutrients-16-04161-t003:** Clinical rheumatological scores related to the first and second groups at t0, t45, and t90.

		First Group				Second Group			
		Average	dev std	*p*-Value	% Change with Respect to t0	Average	dev std	*p* Value	% Change with Respect to t0
WPI	t0	15.46	3.97	5.17 × 10^−9^		14.44	3.22		
	t45	5.70	3.59	4.94 × 10^−2^	−63.15	10.74	5.39	0.0022	−25.67
	t90	7.50	3.85	8.32 × 10^−7^	−51.48	10.50	5.35	0.8910	−25.00
SSS	t0	9.33	1.71	4.08 × 10^−8^		9.44	1.82		
	t45	5.35	2.21	5.71 × 10^−2^	−42.70	7.05	2.52	0.0003	−25.33
	t90	6.23	2.60	1.73 × 10^−5^	−33.27	7.16	2.59	0.7730	−24.15
HAM-A	t0	28.50	8.44	2.57 × 10^0^		28.58	8.70		
	t45	11.18	5.13	1.90 × 10^−1^	−60.76	20.58	10.12	0.0130	−28
	t90	12.68	7.17	6.41 × 10^−9^	−55.50	21.33	8.79	0.5500	−25.37
HAM-D	t0	19.21	5.27	1.19 × 10^−5^		17.89	4.04		
	t45	12.27	6.16	2.73 × 10^−1^	−36.10	14.89	4.02	0.0370	−16.8
	t90	10.14	4.41	5.31 × 10^−7^	−47.22	18.44	4.19	0.0100	−3.17

**Table 4 nutrients-16-04161-t004:** VIP score analysis: metabolites discriminating FM1 at t0, after 45 days (t45) of the VLCKD diet, and after an additional 45 days (t90) of the LOGI diet. Results are based on NMR metabolomic sera, urine, and saliva analysis. The metabolites are ranked according to decreasing VIP score values.

Serum	FM1		
Metabolites Vip > 1.7	t0	t45	t90
Guanidoacetate	*−*	*++*	*+*
ATP	*++*	*+*	*−*
N-acetylaspartate	*+*	*−*	*++*
D-glucoronic acid	*++*	*+*	*−*
Ascorbic acid	*−*	*+*	*++*
Ornithine	*−*	*+*	*++*
Saccaric acid	*−*	*+*	*++*
Phenylanine	*++*	*+*	*−*
Malic acid	*−*	*+*	*++*
β-Hydroxyisovalerate	*−*	*++*	*+*
Glycogen	*++*	*−*	*+*
L-Leucine	*−*	*+*	*++*
Urine	FM1		
Metabolites VIP > 1.7	t0	t45	t90
Valerate	*++*	*−*	*+*
B-hydroxyisovalerate	*++*	*+*	*−*
Isoleucine	*++*	*−*	*+*
Serine	*++*	*−*	*+*
Fucose	*−*	*+*	*−*
Citrulline	*++*	*−*	*+*
TMAO	*−*	*++*	*+*

**Table 5 nutrients-16-04161-t005:** The enrichment tool dysregulated biochemical pathways using the matrix of serum and urine. The table shows the metabolites responsible for dysmetabolism (Hits) and the *p*-value (raw p). The *p*-value was adjusted using Bonferroni correction and the false discovery rate (FDR).

*Serum Pathways*	*Hits*	*Raw p*	*Holm Adjust*	*FDR*	*Impact*
Arginine biosynthesis	8	1.27 × 10^−9^	5.99 × 10^−9^	5.99 × 10^−9^	0.59898
Glycine, serine, and threonine metabolism	8	3.17 × 10^−8^	1.46 × 10^−8^	7.45 × 10^−8^	0.27228
Alanine, aspartate, and glutamate metabolism	11	7.71 × 10^−8^	3.47 × 10^−8^	1.21 × 10^−8^	0.77725
Galactose metabolism	7	8.16 × 10^−6^	3.43 × 10^−6^	6.39 × 10^−6^	0.39113
Ascorbate and aldarate metabolism	3	5.32 × 10^−62^	2.13 × 10^−6^	3.12 × 10^−6^	0.52381
Arginine and proline metabolism	7	1.51 × 10^−48^	5.28 × 10^−4^	5.46 × 10^−4^	0.42441
Beta-alanine metabolism	3	6.02 × 10^−43^	1.87 × 10^−4^	1.66 × 10^−4^	0.45522
Tyrosine metabolism	4	8.94 × 10^−40^	2.68 × 10^−3^	2.34 × 10^−3^	0.44049
Cysteine and methionine metabolism	3	1.30 × 10^−39^	3.76 × 10^−3^	3.21 × 10^−3^	0.16429
Glycerolipid metabolism	2	2.03 × 10^−35^	5.47 × 10^−3^	4.53 × 10^−3^	0.33022
Nicotinate and nicotinamide metabolism	2	1.08 × 10^−32^	2.70 × 10^−3^	2.21 × 10^−3^	0.23465
Fructose and mannose metabolism	3	8.09 × 10^−32^	1.94 × 10^−3^	1.58 × 10^−3^	0.13078
Phenylalanine metabolism	2	3.02 × 10^−23^	5.43 × 10^−2^	4.58 × 10^−2^	0.35714
Starch and sucrose metabolism	3	8.08 × 10^−22^	1.05 × 10^−2^	1.09 × 10^−2^	0.49833
Taurine and hypotaurine metabolism	2	1.55 × 10^−14^	1.08 × 10^−2^	1.77 × 10^−2^	0.82857
** *Urine Pathways* **	** *Hits* **	** *Raw p* **	** *Holm adjust* **	** *FDR* **	** *Impact* **
Alanine, aspartate, and glutamate metabolism	6	4.52 × 10^−10^	2.04 × 10^−9^	7.09 × 10^−19^	0.53446
Glycine, serine, and threonine metabolism	8	3.06 × 10^−9^	1.32 × 10^−6^	2.87 × 10^−15^	0.46212
Phenylalanine, tyrosine, and tryptophan biosynthesis	3	1.45 × 10^−8^	5.93 × 10^−6^	8.50 × 10^−13^	1
Phenylalanine metabolism	3	1.45 × 10^−8^	5.93 × 10^−6^	8.50 × 10^−13^	0.61904
Glutathione metabolism	4	1.11 × 10^−5^	3.90 × 10^−5^	4.03 × 10^−14^	0.28271
Galactose metabolism	7	2.91 × 10^−5^	9.32 × 10^−5^	8.55 × 10^−10^	0.14531
Taurine and hypotaurine metabolism	2	3.48 × 10^−5^	1.08 × 10^−3^	9.62 × 10^−10^	0.82857
Tyrosine metabolism	4	8.67 × 10^−4^	2.51 × 10^−3^	2.14 × 10^−8^	0.44049
Arginine and proline metabolism	6	2.53 × 10^−2^	7.08 × 10^−3^	5.94 × 10^−8^	0.37674
Ascorbate and aldarate metabolism	3	2.45 × 10^−2^	5.87 × 10^−3^	4.79 × 10^−7^	0.52381
Arginine biosynthesis	6	3.37 × 10^−2^	7.74 × 10^−3^	6.33 × 10^−7^	0.48223
Starch and sucrose metabolism	2	1.18 × 10^−2^	1.53 × 10^−3^	1.59 × 10^−2^	0.42527
Beta-alanine metabolism	3	1.78 × 10^−2^	2.85 × 10^−2^	2.62 × 10^−4^	0.39925

## Data Availability

The original contributions presented in this study are included in the article/Appendix A. Further inquiries can be directed to the corresponding author.

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
