# Peer review of "The Effectiveness of the Low-Glycemic and Insulinemic (LOGI) Regimen in Maintaining the Benefits of the VLCKD in Fibromyalgia Patients"

_nutrients, 2024, doi:10.3390/nu16234161_

Round 1

Reviewer 1 Report

Comments and Suggestions for Authors

The manuscript is well-written and understandable.

The research topic is important from the view of conceptual research, and also for clinical practice.

The study sample of ca. 40 individuals is very small.

Has an a priori power analysis been conducted?

Are estimates reliable with such a small sample?

The studied 45-day period is nice.

However, more detailed description regarding drop-out is needed.

Likewise, the satisfaction and motivation during the long follow-up would be interesting to see.

For translation into daily health behavior, is it likely that patients are able to appropriately follow the nutritional pattern without the experimenter checking?

The detailed physiological mechanisms underlying the metabolic imbalances in the context of fibromyalgia could be elaborated in more depth.

Can the authors exclude negative side effects of the treatment?

Is the effect comparable in women and men?

The practical relevance for clinical work could be highlighted in more details.

What is the cost-effectiveness ratio compared to the alternative approaches?

Author Response

The research topic is important from the view of conceptual research, and also for clinical practice.

1-The study sample of ca. 40 individuals is very small. Has an a priori power analysis been conducted? Are estimates reliable with such a small sample?
We reported Power analysis in section 2.1 Participants and samples size calculation lines 129-137
2- The studied 45-day period is nice.However, more detailed description regarding drop-out is needed. 
 Reasons of drop-out in FM2 patients can be summarized as follows : 
1) The LOGI Diet is not able to determine a reduction in hunger, like the Ketogenic Diet. In this case, an anorectic effect occurs due to the beta-hydroxybutyrate high concentration. 
2) The LOGI Diet, compared to the Ketogenic Diet, demonstrated minimal effects against the fibromyalgia symptoms, in particular tiredness and widespread tenderness. Moreover, it is to be considered the peculiarity of the fibromyalgia patient who is often unable to follow therapies as disappointed from previous therapeutic failures.
Likewise, the satisfaction and motivation during the long follow-up would be interesting to see. For translation into daily health behavior, is it likely that patients can appropriately follow the nutritional pattern without the experimenter checking?
We thank the reviewer for the observation. Indeed, our follow-up was based on an interview every 15 days. Moreover, telephone communication between the patients and the investigators was constantly possible. All together these observational data show that long-term satisfaction is higher in the FM1 group. 
3-The detailed physiological mechanisms underlying the metabolic imbalances in the context of fibromyalgia could be elaborated in more depth.
 We thank the reviewer for his comment we add this information in lines 60-65
4-Can the authors exclude negative side effects of the treatment?
Patients were monitored regularly by both rheumatologists and nutrition experts to avoid adverse events. Blood and urine standard biochemical parameters were regularly controlled, and no out-of-range parameters were observed.
5-Is the effect comparable in women and men?
It was not possible to make a women/men comparison. As fibromyalgia is dramatically more common among women,  we were unable to recruit male patients.
6-The practical relevance for clinical work could be highlighted in more details.
Fibromyalgia currently lacks appropriate diagnosis and effective therapy. Our metabolomics analysis supports the effectiveness of a diet therapy approach in managing the effects of a pathology that is widespread mainly and has a significant socio-economical impact.
7-What is the cost-effectiveness ratio compared to the alternative approaches?
The therapeutic proposal has a great cost-effective ratio, not only due to the modest expenses of the nutritional therapy but also because successful fibromyalgia management reduces the use of diagnostic procedures and pharmacological treatments

Reviewer 2 Report

Comments and Suggestions for Authors In the current article the authors investigated the dietary interventions for fibromyalgia by evaluating the impact of transitioning from a very-low-calorie ketogenic diet to a low-glycaemic insulinemic diet over an additional 45-day period, from day 45 to day 90. Their findings underline that while Oloproteic Diet serves as an effective initial intervention for correcting metabolic imbalances and alleviating fibromyalgia symptoms, transitioning to a low-glycaemic insulinemic diet offers a practical and sustainable dietary strategy. In my opinion the study is interesting and welcome.

Some suggestions:

1. Lines 57-60: Add please some details concerning the following statement: “Recent studies suggest a potential link between FM and systemic metabolic dysfunctions with a shared inflammatory origin.[8-12] As a proof of concept, we previously demonstrated that the metabolome of fibromyalgia patients exhibits an abnormal concentration of several metabolites associated with systemic inflammatory conditions. [5,13]”

2. Lines 85-87, you wrote: “Based on NMR-based metabolomic data, changes induced by the Oloproteic treatment were evident in all the metabolic pathways related to fibromyalgia dysfunction. [13]”. Please add which  are “all the metabolic pathways”.

3. In my opinion the place of the information presented in lines 106-127 is not at the introduction.

4. The eligibility criteria (inclusion/exclusion criteria) must be presented in the article not as Supplementary information.

5. line 137- It is better to write biochemical evaluation not hematological evaluation.

6. Table S2. Please specify more clearly which parameters were determined from blood and which from urine.

7. line 143- Give please some details regarding the statement “Clinical and laboratory evaluation was performed as previously reported. [12,13,16]”

8. At material and method you must describe in detail in what do the two diets consist of. At this moment the LOGI diet is partially described and the oloproteic diet is not described.

9. At lines 106-108 you wrote: “By employing a combination of clinical evaluations and non-invasive NMR metabolomic analyses of serum, urine, and saliva samples, we monitored the physiological and biochemical changes in the participants from day 45 to day 90” . At point 2.5 Sample preparation for NMR metabolomic analysis you described only the serum and urine preparation. What about saliva samples preparation? Please clarify.

10. Table 2 is not correctly positioned in the article.

11.Improve please the quality of Figures 1 and 3.

12. Give please details concerning the following two statements:

- lines 355-57: “As previously demonstrated, the rebalance of the neurotransmission amino acid metabolism explains an improvement of fibromyalgia symptoms such as neuropathic and muscular pain. [35,36]”

-lines 366-369: “A significant decrease in dysbiosis biomarkers in urine (including hydroxy valerate, 366 valerate, citrulline, and TMAO) and saliva (glycolate and urea) [39,40](Tables 3,4; Tables 367 S2, S3) is confirmed in the t45-t90 time range for FM1 patients. Reduction of these metab-368 olites is associated with improving inflammatory conditions in the gut.

Author Response

we attached the rebuttal
